# SparseSwaps: Tractable LLM Pruning Mask Refinement at Scale

## Abstract

The resource requirements of Neural Networks can be significantly reduced through pruning – the removal of seemingly less important parameters. However, with the rise of Large Language Models (LLMs), full retraining to recover pruning-induced performance degradation is often prohibitive and classical approaches such as global magnitude pruning are suboptimal on Transformer architectures. State-of-the-art methods hence solve a layer-wise *mask selection problem*, the problem of finding a pruning mask which minimizes the per-layer pruning error on a small set of calibration data. Exactly solving this problem to optimality using Integer Programming (IP) solvers is computationally infeasible due to its combinatorial nature and the size of the search space, and existing approaches therefore rely on approximations or heuristics. In this work, we demonstrate that the mask selection problem can be made drastically more tractable at LLM scale. To that end, we decouple the rows by enforcing equal sparsity levels per row. This allows us to derive optimal *1-swaps* (exchanging one kept and one pruned weight) that can be computed efficiently using the Gram matrix of the calibration data. Using these observations, we propose a tractable and simple 1-swap algorithm that warm starts from any pruning mask, runs efficiently on GPUs at LLM scale, and is essentially hyperparameter-free. We demonstrate that our approach reduces per-layer pruning error by up to 60% over Wanda (Sun et al., 2023) and consistently improves perplexity and zero-shot accuracy across state-of-the-art GPT architectures.

## 1 Introduction

*Pruning after training* (Han et al., 2015; Gale et al., 2019; Lin et al., 2020; Hoefler et al., 2021; Zimmer et al., 2025) is a state-of-the-art technique to reduce the resource requirements of neural networks. A simple yet effective approach to obtain such *sparse* models starts from a pretrained *dense* model, removes seemingly unimportant parameters based on their magnitude, and requires retraining to compensate for pruning-induced performance degradation. However, while the inexpensive, data-free magnitude criterion has often achieved strong performance on traditional architectures (Gale et al., 2019; Zimmer et al., 2023b), pruning has undergone a paradigm shift with the rise of large pretrained foundation models, particularly LLMs.

First, the size of the models has shifted the focus toward retraining-free pruning criteria, as retraining is often computationally expensive if not infeasible, with parameter-efficient fine-tuning (Lialin et al., 2023; Zimmer et al., 2023a) being an exception. Secondly, systematic activation outliers (Dettmers et al., 2022) and highly important *super-weights* (Yu et al., 2025) in sufficiently large *Transformers* (Vaswani et al., 2017) have rendered magnitude pruning no better than random pruning for LLMs (Sun et al., 2023; Yin et al., 2023). Lastly, state-of-the-art methods (Frantar & Alistarh, 2023; Sun et al., 2023; Zhang et al., 2024) prune *layer-wise*: they split the pruning problem into per-layer subproblems, pruning layers sequentially and independently using a small calibration dataset to estimate parameter importance. Rather than optimizing the *global* loss, such approaches minimize a per-layer *local* pruning loss. Specifically, for a single layer with calibration input matrix $X \in \mathbb{R}^{d_{in} \times B}$ and weights $W \in \mathbb{R}^{d_{out} \times d_{in}}$, the objective becomes

$$\min_M \|WX - (M \odot W)X\|_F^2, \tag{1}$$

where $M \in \{0,1\}^{d_{out} \times d_{in}}$ is a binary pruning mask achieving a desired level of sparsity, e.g., $\|M\|_0 \leq k$ for unstructured sparsity, and $\odot$ denotes the element-wise multiplication or Hadamard product. Here, $B = N \cdot L$ with $N$ being the number of samples in the calibration batch and $L$ being the sequence length.

Solving this combinatorial *mask selection problem* to optimality is NP-hard due to feature correlations: selecting $k$ of $d_{out} \cdot d_{in}$ weights yields a cardinality-constrained binary quadratic program (a best-subset selection variant). Even for a single row $i$, the problem reduces to

$$\min_{m_i} \left\| w_i^\top X - (m_i \odot w_i)^\top X \right\|_F^2 = \min_{m_i} \sum_{k=1}^{B} \left( \sum_{j=1}^{d_{in}} (1 - m_{ij}) w_{ij} X_{jk} \right)^2,$$

where $w_i \in \mathbb{R}^{d_{in}}$ and $m_i \in \{0,1\}^{d_{in}}$ denote the $i$-th row of $W$ and $M$, respectively. While IP solvers could theoretically provide optimal solutions, the combinatorial search over mask entries makes this infeasible for LLMs. In practice, existing methods therefore relax Equation 1 or approximate it.

However, with deployed LLMs now serving millions of users, it becomes increasingly worthwhile to invest substantial resources to obtain pruned models that reach high performance, because the pruning cost is paid once during training whereas inference costs scale with the number of requests. In this work, we revisit the per-layer mask selection problem and demonstrate that it can be operationalized at LLM scale, enabling monotone improvements with each optimization step rather than relying on proxy importance scores. To that end, we observe that enforcing equal sparsity-level across rows ensures *row-wise separability* that yields independent objectives. This makes the problem drastically more tractable and leads to good practical performance for LLMs. Instead of trying to obtain exact solutions via IP solvers, we instead propose a GPU-accelerated local optimization algorithm based on *1-swaps* (exchanging one kept and one pruned weight) that perform *exact and efficient local refinement with incremental cost updates* using the Gram matrix $G = XX^\top$ to monotonically decrease the objective from any warm start.

The resulting method, which we term SparseSwaps, can start from any warm-start mask, evaluates the exact per-row quadratic loss, and is scalable, parallelizable across rows, almost hyperparameter-free, and deterministic for a fixed warm start. With only few 1-swap iterations, it can reduce the per-layer pruning error by up to 60% compared to Wanda and improves final perplexity and zero-shot accuracy across architectures. Our approach is a post-hoc refinement of existing pruning methods that can significantly improve upon the state of the art for unstructured, per-row, or $N:M$ sparsity.

**Contributions.** Our contributions are as follows:

1. **Making the Mask Selection problem tractable.** We observe that a) enforcing equal sparsity levels per row decouples the rows, and that b) optimal 1-swaps (exchanging one kept and one pruned weight) can be evaluated efficiently using the Gram matrix $G = XX^\top$ of the calibration data, ensuring efficient lookups when determining the most beneficial swap.

2. **SparseSwaps: a practical post-hoc pruning algorithm.** Building on these observations, we propose SparseSwaps, a plug-and-play 1-swap refinement that starts from any warm-start mask and monotonically decreases the exact per-row objective under per-row or $N:M$ constraints. In particular, SparseSwaps is almost hyperparameter-free, completely parallelizable across rows and scalable to LLMs.

3. **Computational study.** We verify our hypotheses on state-of-the-art Generative Pretrained Transformer (GPT) architectures and demonstrate that SparseSwaps delivers large reductions in local pruning error (up to 60% per-layer error reduction over Wanda) and strong perplexity and zero-shot gains across a wide range of different LLMs. We conduct a series of ablations highlighting the advantages and drawbacks of the proposed approach.

**Further related work.** *Post-training pruning* has a long history, and while *magnitude pruning* (Janowsky, 1989; Han et al., 2015) is among the most popular criteria, it is not the only one (cf. LeCun et al., 1989; Hassibi & Stork, 1993; Molchanov et al., 2016; Yeom et al., 2019); see Hoefler et al. (2021) for a comprehensive review. Despite their simplicity, magnitude-based methods

have been shown to produce sparse models competitive with far more complex algorithms for convolutional architectures (Gale et al., 2019; Zimmer et al., 2023b). For LLMs, however, magnitude pruning is argued to be unsuitable (Yin et al., 2023). Consequently, there is growing interest in criteria beyond magnitude that achieve high performance on LLMs, and do so without requiring an expensive retraining procedure (Kwon et al., 2022; Frantar & Alistarh, 2023; Sun et al., 2023; Zhang et al., 2024). In this work, we develop a post-hoc refinement of existing methods, rather than proposing a new criterion. A related approach, DSnoT (Zhang et al., 2023), also performs iterative weight swaps but differs significantly in its optimization strategy. Inspired by *dynamic sparse training* (cf. Evci et al., 2020), DSnoT prunes and regrows weights based on expected reconstruction-error improvements, using feature means and variances as surrogates. While effective, it does not guarantee a monotonic decrease in the true pruning error, whereas our method does. We compare the two empirically and find that SparseSwaps consistently outperforms DSnoT.

*Subset selection and IP approaches.* To solve Equation 1 to global optimality, which can be formulated as a *mixed-integer nonlinear program (MINLP)*, several efficient open-source solvers are available, including SCIP (Bolusani et al., 2024), Bonmin (Bonami et al., 2008), and SHOT (Lundell et al., 2022), among others. In particular, the recently introduced Boscia solver (Hendrych et al., 2025) is particularly well-suited, as it exploits the problem's combinatorial structure. While we demonstrate how the problem can be made drastically more tractable, explicit solution remains very time-consuming for large instances; we therefore opt for a GPU-friendly 1-swap approach that avoids moving large tensors to the CPU for IP solvers. We leave such an extension for future work.

## 2 METHODOLOGY

In the following, we use uppercase letters for matrices ($W$, $X$, $M$) and lowercase letters for scalars and vectors. Matrix entries are denoted $W_{ij}$ for the element in row $i$, column $j$. Rows of matrices are denoted with lowercase subscripts: $w_i$ represents the $i$-th row of matrix $W$. Row and column slices use colon notation: $X_{j,:}$ for the $j$-th row and $X_{:,k}$ for the $k$-th column. We use $\odot$ for element-wise multiplication, $\|\cdot\|_F$ for Frobenius norm, and $\|\cdot\|_2$ for $\ell_2$ norm.

### 2.1 PRELIMINARIES

Before describing our proposed method, we make several assumptions and observations that make the problem tractable.

#### 2.1.1 EQUAL SPARSITY-LEVEL ACROSS ROWS DOES NOT NEED TO BE DETRIMENTAL

First, note that the objective in Equation 1 decomposes into a sum of $d_{out}$ row-wise quadratics,

$$\|WX - (M \odot W)X\|_F^2 = \sum_{i=1}^{d_{out}} \left\|w_i^\top X - (m_i \odot w_i)^\top X\right\|_2^2$$

where $w_i \in \mathbb{R}^{d_{in}}$ and $m_i \in \{0,1\}^{d_{in}}$ denote the $i$-th row of $W$ and $M$, respectively. This alone does not make the corresponding minimization problem row-separable under unstructured sparsity, since the matrix cardinality constraint couples rows. In contrast, semi-structured patterns like per-row sparsity (keep $k$ per row) or $N{:}M$ (prune $M{-}N$ per block of $M$ weights) enforce equal per-row sparsity and fully decouples rows which can now be solved independently. We therefore focus on the decoupled case, allowing to treat each row separately and reducing the problem to

$$\min_{m_i} \left\|w_i^\top X - (m_i \odot w_i)^\top X\right\|_F^2 = \min_{m_i} \sum_{k=1}^{B} \left(\sum_{j=1}^{d_{in}} (1 - m_{ij})w_{ij}X_{jk}\right)^2 \tag{2}$$

for each row $i \in \{1, \ldots, d_{out}\}$. Note that, for LLMs, Sun et al. (2023) observe that row-wise sparsity benefits performance for both Wanda and magnitude pruning. We therefore argue that enforcing per-row sparsity rather than unstructured sparsity is justified and need not harm final performance, at least for LLMs. For semi-structured sparsity, the rows are decoupled anyway.

As a side note, since positive scaling preserves minima and by applying Jensen's inequality, one can now easily derive the Wanda criterion:

$$\min_{m_i} \sum_{k=1}^{B} \left( \sum_{j=1}^{d_{in}} (1 - m_{ij}) w_{ij} X_{jk} \right)^2 \leq \min_{m_i} \sum_{k=1}^{B} \left( \sum_{j=1}^{d_{in}} (1 - m_{ij})^2 w_{ij}^2 X_{jk}^2 \right)$$

$$= \min_{m_i} \sum_{j=1}^{d_{in}} (1 - m_{ij})^2 w_{ij}^2 \|X_{j,:}\|_2^2. \tag{3}$$

Equation 3 is solved by pruning entries with the smallest saliency $|w_{ij}| \cdot \|X_{j,:}\|_2$, i.e., precisely the Wanda criterion. Thus, Wanda optimizes an upper bound to the original problem that ignores within-row interactions, making the combinatorial problem tractable.

### 2.1.2 AVOIDING INTERMEDIATE VALUE CACHING THROUGH THE GRAM MATRIX FORMULATION

Naively caching all $B \cdot d_{in}$ intermediate products $w_{ij} X_{jk}$ in Equation 2 to evaluate candidate masks is prohibitive. To illustrate the scale, consider a single row of the largest matrix in a LLAMA-2-7B Transformer block: the up_proj matrix with input dimension $d_{in} = 4096$. With $N = 128$ samples and sequence length $L = 4096$ (so $B = N \cdot L = 524{,}288$), caching all products $w_{ij} X_{jk}$ for that row requires $524{,}288 \times 4096 \approx 2.15$ billion float32 values (about 8.6GB); across all 11,008 rows this totals about 94.6TB.

A straightforward way to circumvent this issue is to consider a single row and derive a compact formulation of the per-row loss through the Gram matrix $G = XX^\top \in \mathbb{R}^{d_{in} \times d_{in}}$. For notational convenience, we drop the row index $i$ throughout the remainder of this section and write $w \in \mathbb{R}^{d_{in}}$ for the row's weight vector and $m \in \{0,1\}^{d_{in}}$ for its mask. The per-row loss from Equation 2 is

$$L := \left\| w^\top X - (m \odot w)^\top X \right\|_F^2 = \left\| (w - m \odot w)^\top X \right\|_F^2 = (w - m \odot w)^\top G (w - m \odot w).$$

Hence, the loss depends on $X$ only through the Gram matrix $G$, which can be accumulated on-the-fly as calibration samples pass through the layer: $G = \sum_{b=1}^{B} X_{:,b} X_{:,b}^\top$. Unlike the per-row formulation in the introduction, which would require caching all $B \cdot d_{in}$ intermediate products $w_j X_{jk}$, we only need to maintain the $d_{in} \times d_{in}$ matrix $G$ – a reduction from $\mathcal{O}(B \cdot d_{in})$ to $\mathcal{O}(d_{in}^2)$, with $d_{in}$ typically being much smaller than $B$.

**Remark 1.** *A different (but in practice slightly less efficient) perspective on this reduction is through the* unitary invariance *of the Frobenius norm used in our pruning objective: for any matrix $A$ and unitary matrix $U$ (i.e., $U^{-1} = U^\top$), we have $\|AU\|_F = \|A\|_F$. This property enables significant computational savings through Singular Value Decomposition (SVD) compression. Precisely, let $X = U\Sigma V^\top$ be the SVD of calibration data $X \in \mathbb{R}^{d_{in} \times B}$. Since $B > d_{in}$, we can write $\Sigma = [\Sigma' \mid 0]$ with $\Sigma' \in \mathbb{R}^{d_{in} \times d_{in}}$ containing the singular values on its diagonal. The compressed representation is simply $X' = U\Sigma' \in \mathbb{R}^{d_{in} \times d_{in}}$. Letting $w_p = w - m \odot w$ for brevity, the key insight is that pruning decisions remain equivalent under this compression:*

$$\|w_p X\|_F^2 = \left\| w_p U\Sigma V^\top \right\|_F^2 = \|w_p U\Sigma\|_F^2 = \|w_p U[\Sigma' \mid 0]\|_F^2 = \|w_p U\Sigma'\|_F^2 = \|w_p X'\|_F^2,$$

*where we used unitary invariance w.r.t. $V$ and that the zero columns do not contribute to the Frobenius norm. Equivalently, we have*

$$X'X'^\top = U\Sigma'\Sigma'^\top U^\top = U\Sigma\Sigma^\top U^\top = XX^\top = G,$$

*since $\Sigma\Sigma^\top = \Sigma'\Sigma'^\top$ (the zero columns of $\Sigma$ do not contribute). Since all subsequent operations depend solely on $G$, we accumulate $G$ directly during calibration and avoid the SVD entirely.*

### 2.1.3 EFFICIENT 1-SWAP EVALUATION THROUGH EFFICIENT COST LOOKUPS AND UPDATES

While the global mask selection problem is NP-hard, we can still make efficient progress via local search. Starting from any feasible mask $m \in \{0,1\}^{d_{in}}$, the idea is to iteratively perform 1-swaps that exchange one kept and one pruned weight to reduce $L$ while preserving the sparsity level. The key observation is that each candidate swap can be evaluated in $\mathcal{O}(1)$ time using $G$ and an auxiliary

*correlation vector c.* To that end, let $\mathcal{P} = \{j : m_j = 0\}$ denote the set of currently pruned weight indices and analogously $\mathcal{U} = \{j : m_j = 1\}$ denote the set of unpruned (kept) weight indices. Letting further $\phi_j = X_{j,:}^\top \in \mathbb{R}^B$ denote the $j$-th row (or feature vector)of $X$, we can write

$$(w - m \odot w)^\top X = \sum_{j=1}^{d_{in}} (1 - m_j) w_j X_{j,:} = \sum_{j \in \mathcal{P}} w_j \phi_j^\top = r^\top,$$

where we define the *reconstruction residual* $r = \sum_{j \in \mathcal{P}} w_j \phi_j \in \mathbb{R}^B$, the total contribution of all pruned weights to the layer output. Hence, clearly, the loss is $L = \|r\|_2^2 = r^\top r$.

We define the *correlation vector* $c = (c_1, \ldots, c_{d_{in}})^\top \in \mathbb{R}^{d_{in}}$ with entries

$$c_i = \langle \phi_i, r \rangle = \langle \phi_i, \sum_{j \in \mathcal{P}} w_j \phi_j \rangle = \sum_{j \in \mathcal{P}} w_j \langle \phi_i, \phi_j \rangle = \sum_{j \in \mathcal{P}} w_j G_{ij},$$

which measures how each feature $\phi_i$ correlates with the current residual. In vector form, $c = G \cdot ((\mathbb{1} - m) \odot w)$.

**Swap cost formula.** A 1-swap removes index $u \in \mathcal{U}$ from the unpruned set (making it pruned) and adds index $p \in \mathcal{P}$ to the unpruned set (making it unpruned). The new residual is $r' = r + w_u \phi_u - w_p \phi_p$, and the change in loss is

$$\Delta L_{u,p} = \|r'\|_2^2 - \|r\|_2^2 = \|r + w_u \phi_u - w_p \phi_p\|_2^2 - \|r\|_2^2$$
$$= 2w_u \langle \phi_u, r \rangle + w_u^2 \|\phi_u\|_2^2 - 2w_p \langle \phi_p, r \rangle + w_p^2 \|\phi_p\|_2^2 - 2w_u w_p \langle \phi_u, \phi_p \rangle.$$

Using $c_i = \langle \phi_i, r \rangle$ and $G_{ij} = \langle \phi_i, \phi_j \rangle$, this simplifies to

$$\Delta L_{u,p} = 2w_u c_u + w_u^2 G_{uu} - 2w_p c_p + w_p^2 G_{pp} - 2w_u w_p G_{up}. \tag{4}$$

Given the precomputed Gram matrix $G$ and correlation vector $c$, each swap evaluation requires only *scalar lookups*. Evaluating all $|\mathcal{U}| \cdot |\mathcal{P}|$ possible swaps therefore costs $\mathcal{O}(|\mathcal{U}| \cdot |\mathcal{P}|)$ total. By systematically testing all $(d_{in} - |\mathcal{P}|) \cdot |\mathcal{P}|$ possible 1-swap operations (adding one of $|\mathcal{U}| = d_{in} - |\mathcal{P}|$ unpruned weights to $\mathcal{P}$, removing one of $|\mathcal{P}|$ pruned weights from $\mathcal{P}$) evaluating the improvement using the above expression, we iteratively pick a best swap and update the mask until we have reached a satisfactory solution or one optimal w.r.t. 1-swap operations. The only issue that remains is to update the correlation vector after each swap.

**Correlation vector update.** After accepting a swap $(u^*, p^*)$, the residual changes to $r' = r + w_{u^*} \phi_{u^*} - w_{p^*} \phi_{p^*}$. The correlation vector updates as

$$c_i \leftarrow c_i + w_{u^*} G_{i,u^*} - w_{p^*} G_{i,p^*}, \tag{5}$$

or in vector form, $c \leftarrow c + w_{u^*} G_{:,u^*} - w_{p^*} G_{:,p^*}$. This only requires accessing two columns of $G$ and costs $\mathcal{O}(d_{in})$.

**Why picking $p$ and $u$ separately is suboptimal.** The interaction term $-2w_u w_p G_{up}$ in Equation 4 shows that the best $u$ depends on the chosen $p$ (and vice versa). Consequently, selecting $p$ and $u$ based on their individual effects can yield a detrimental swap, as the following example for the scalar case with $B = 1$ and $d_{in} = 4$ shows. Let the current pruned weight contributions be $\{+10, -1\}$, so $r = 9$ and $L = 81$, and let the unpruned weight contributions be $\{+9, -9\}$. The best 1-swap is to unprune the $-1$ contribution and prune the $-9$ contribution, giving $r' = 10 + (-9) = 1$ and $L' = 1$. However, if we instead greedily remove the best $p$ in isolation, we unprune $+10$ since $(9 - 10)^2 = 1$ is minimal. We must then add one index; the best addition in isolation to the original pruned-weight contributions $\{+10, -1\}$ is $-9$. In combination, the greedily chosen swap leads to $r' = -1 + (-9) = -10$ and $L' = 100$ – *worse* than the starting point. The error stems precisely from ignoring the interaction term when selecting $(p, u)$.

## 2.2 THE SPARSESWAPS ALGORITHM

Building upon the preceding observations, we present our complete algorithm. The method takes as input a weight matrix $W \in \mathbb{R}^{d_{out} \times d_{in}}$, the Gram matrix $G = XX^\top \in \mathbb{R}^{d_{in} \times d_{in}}$ (accumulated

during calibration), and a warmstart pruning mask $M^{\text{init}} \in \{0,1\}^{d_{out} \times d_{in}}$ that already satisfies the desired sparsity constraints, e.g., obtained from Wanda or RIA (Zhang et al., 2024).

The algorithm enforces any sparsity pattern that operates per-row, including per-row sparsity (fixed number of zeros per row, cf. Sun et al. (2023)) and structured $N{:}M$ sparsity patterns (e.g., 2:4 or 4:8, Mishra et al. (2021)). All swap operations maintain the sparsity constraints throughout optimization; for $N{:}M$ sparsity, swaps are restricted to occur only within the same $N{:}M$ blocks, while for per-row sparsity, the total number of pruned weights per row remains constant. Even though each swap only changes two mask entries, the cumulative effect of multiple swaps can dramatically reduce reconstruction error compared to the initial solution.

---

**Algorithm 1** SparseSwaps: 1-Swap Pruning Optimization

---

**Require:** $W \in \mathbb{R}^{d_{out} \times d_{in}}$, Gram matrix $G = XX^\top \in \mathbb{R}^{d_{in} \times d_{in}}$, warmstart mask $M^{\text{init}}$, $T_{\max}$
**Ensure:** Improved pruning mask $M$
1: $M \leftarrow M^{\text{init}}$      ▷ Initialize with warmstart solution
2: **for** $i = 1$ to $d_{out}$ **do**      ▷ Process each row independently
3:     $w \leftarrow W_{i,:}, m \leftarrow M_{i,:}$      ▷ Extract row weights and mask
4:     $\mathcal{P} \leftarrow \{j : m_j = 0\}, \mathcal{U} \leftarrow \{j : m_j = 1\}$      ▷ Pruned and unpruned sets
5:     $c \leftarrow G \cdot ((\mathbb{1} - m) \odot w)$      ▷ Initialize correlation vector
6:     **for** $t = 1$ to $T_{\max}$ **do**
7:        $(p^*, u^*) \leftarrow \arg\min_{(p,u)} \Delta L_{u,p}$      ▷ Best swap via Equation 4
8:        **if** $\Delta L_{u^*,p^*} < 0$ **then**      ▷ Swap improves objective
9:          $m_{p^*} \leftarrow 1, m_{u^*} \leftarrow 0$      ▷ Perform swap
10:          $\mathcal{P} \leftarrow (\mathcal{P} \setminus \{p^*\}) \cup \{u^*\}, \mathcal{U} \leftarrow (\mathcal{U} \setminus \{u^*\}) \cup \{p^*\}$
11:          $c \leftarrow c + w_{u^*} G_{:,u^*} - w_{p^*} G_{:,p^*}$      ▷ Update correlation vector
12:        **else**
13:          **break**      ▷ Local optimum reached
14:        **end if**
15:     **end for**
16:     $M_{i,:} \leftarrow m$      ▷ Store optimized row
17: **end for**

---

We explain the main phases of the algorithm:

**Preparation:** We initialize with the warmstart mask $M^{\text{init}}$. The Gram matrix $G$ is precomputed once per layer by accumulating $G = \sum_b X_{:,b} X_{:,b}^\top$ during the calibration forward pass.

**Row processing (Lines 2-5):** For each row $i$, we extract weights $w$ and current mask $m$, define pruned and unpruned index sets $\mathcal{P}$ and $\mathcal{U}$, and compute the initial correlation vector $c = G \cdot ((\mathbb{1} - m) \odot w)$.

**1-Swap optimization (Lines 6-15):** We iteratively find the swap $(p^*, u^*)$ minimizing $\Delta L_{u,p}$ (cf. Equation 4) among feasible pairs, evaluating each candidate in $\mathcal{O}(1)$ time. If $\Delta L_{u^*,p^*} < 0$, we accept the swap and update the correlation vector via Equation 5; otherwise we terminate. At all times, the swaps are appropriately constrained: per-row sparsity allows any swap maintaining $|\mathcal{P}|$ constant, while $N{:}M$ sparsity restricts swaps to within the same $N{:}M$ blocks.

**Computational complexity:** The algorithm has complexity $\mathcal{O}(d_{out} \cdot T_{\max} \cdot (|\mathcal{P}| \cdot |\mathcal{U}| + d_{in}))$ per layer, where $T_{\max}$ is the maximum number of swap iterations per row. The $|\mathcal{P}| \cdot |\mathcal{U}|$ term comes from evaluating all candidate swaps (each in $\mathcal{O}(1)$ time via Equation 4), and the $d_{in}$ term from the correlation vector update (Equation 5).

In practice, several factors further reduce runtime. First, we find that even setting $T_{\max} = 1$ or $T_{\max} = 2$ can drastically reduce the local pruning error; values around $T_{\max} = 25$ often suffice to significantly lower model perplexity, with diminishing returns beyond $T_{\max} = 100$. Second, row-wise processing can be batched and vectorized, enabling parallel swap cost computations and mask updates, and rows can be distributed across GPUs if needed. Third, the Gram matrix $G$ is computed once per layer and shared across all rows, and several summands of Equation 4 can be similarly precomputed once per layer.

# 3 EXPERIMENTAL RESULTS

We outline our general experimental approach, detailing datasets, architectures, and metrics. To enable reproducibility, our code will be publicly released. Our study focuses on language modeling within Natural Language Processing (NLP). We use pretrained models from HuggingFace (Wolf et al., 2020), specifically LLAMA-3.1-8B (Grattafiori et al., 2024), GEMMA-2-9B (Riviere et al., 2024), YI-1.5-9B (Young et al., 2025), DEEPSEEK-7B-BASE (Bi et al., 2024), and QWEN2.5-7B (Yang et al., 2025). For calibration, we randomly draw sequences of 2048 tokens from the *C4* dataset (Raffel et al., 2020). For validation, we similarly pick 100 sequences from the validation split. The model performance is assessed via perplexity on the *WikiText* dataset (Merity et al., 2016) and zero-shot accuracy on the EleutherAI evaluation set (Gao et al., 2023). Following Sun et al. (2023), we prune all linear layers, excluding the embedding and final linear head, with uniform sparsity allocation across layers. We provide experiments for unstructured and semi-structured sparsity patterns (Mishra et al., 2021). We use multiple random seeds throughout our experiments.

## 3.1 MASK REFINEMENT AT SCALE

We begin by verifying the effectiveness of SparseSwaps. We make the following observations:

**SparseSwaps consistently improves state-of-the-art methods.** Table 2 summarizes the main results and reports perplexity (upper half, lower is better) and zero-shot accuracy (lower half, higher is better) for warmstart masks (Wanda, RIA) as well as their refinements using DSnoT and SparseSwaps. For both 60% unstructured and 2:4 semi-structured sparsity, SparseSwaps (with 100 1-swap iterations) consistently reduces perplexity and improves zero-shot accuracy over Wanda and RIA warm start masks. While DSnoT similarly yields improvements, it falls short of SparseSwaps. Note that we left the pruning criterion of DSnoT, which partially uses the Wanda saliency, unchanged, even when using RIA warmstart. For unstructured RIA, we report results when enforcing a per-row sparsity constraint; while RIA yields good (and slightly better) results when enforcing truely unstructured sparsity, we decided to include the results for the per-row setting as this allows direct refinement of the mask with SparseSwaps and DSnoT.

**SparseSwaps successfully optimizes the per-layer pruning loss.** Figure 1 shows the per-layer reductions in local pruning error relative to a Wanda Warmstart, grouping layers by their corresponding Transformer block of LLAMA-3.1-8B. We observe drastic improvements of close to 70% compared to Wanda, demonstrating that SparseSwaps is able to successfully optimize the local loss. The `attn.o_proj` seems to consistently benefit the most across blocks, with reductions of the objective in Equation 1 ranging between 40%-60%.

**Large local error reductions do not always imply reduced perplexity.** From Table 2 we observe substantial perplexity gains, especially when sparsity more strongly degrades model quality (cf. Table 4 in the appendix, which shows more drastic improvements when using magnitude pruning, which more strongly degrades model quality). In contrast, when quality is less affected (e.g., at 50% sparsity where Wanda performs well), SparseSwaps yields limited perplexity gains despite significant local error reductions: Table 1 reports perplexity and average relative error reduction (%) versus the number of 1-swap iterations. Zero iterations correspond to the Wanda warm start; one or more iterations correspond to SparseSwaps from Wanda. At 50% sparsity, a single 1-swap iteration lowers relative error by 6.34%, and 200 iterations by nearly 40%, yet perplexity does not improve, but rather slightly increases. This suggests further reducing local error can overfit the calibration data and may not translate to better perplexity, although we note that the perplexity increase is relatively small. These results emphasize that while the reduction of local error is a useful proxy for perplexity reduction when pruning has a higher negative impact on the model, the local error of Equation 1 remains an approximation to the reconstruction error of the entire model.

## 3.2 EFFICIENCY AND HYPERPARAMETER ABLATIONS

**Resource requirements.** SparseSwaps is more resource-intensive than DSnoT and, as a drop-in refinement, requires at least the resources of the chosen warm-start method. Beyond that, SparseSwaps needs memory to store the Gram matrix $G \in \mathbb{R}^{d_{in} \times d_{in}}$ (once per layer) and the correlation vector $c \in \mathbb{R}^{d_{in}}$ (per row), and compute to perform the 1-swaps; see the preceding section for the theoretical complexity. While we have argued in the introduction that additional compute can be

Table 1: LLAMA-3.1-8B: Perplexity (↓) and mean relative reduction in pruning error (↑) versus number of 1-swap iterations for 50% and 60% unstructured sparsity using Wanda warmstart.

| Sparsity | Metric | Number of 1-swap iterations | | | | | | | | |
|---|---|---|---|---|---|---|---|---|---|---|
| | | 0 | 1 | 2 | 5 | 10 | 25 | 50 | 100 | 200 |
| **50%** | Avg. rel. error reduction (%) | 0.00 | 6.34 | 8.77 | 12.51 | 16.38 | 23.52 | 30.04 | 36.48 | 38.95 |
| | Perplexity | 10.13 | 10.31 | 10.40 | 10.41 | 10.39 | 10.38 | 10.27 | 10.30 | 10.34 |
| **60%** | Avg. rel. error reduction (%) | 0.00 | 8.04 | 11.04 | 15.34 | 19.64 | 26.92 | 33.58 | 39.99 | 43.74 |
| | Perplexity | 21.52 | 21.26 | 21.51 | 21.17 | 21.01 | 20.38 | 19.74 | 18.96 | 19.17 |

Table 2: Perplexity (↓, lower is better) and zero-shot accuracy (↑, higher is better) comparison on WikiText and EleutherAI evaluation set. We report DSnoT and SparseSwaps refinement with Wanda and RIA warmstart for unstructured 60% sparsity and semi-structured 2:4 sparsity. Best values are highlighted in **bold**. We omit standard deviations for legibility.

| Perplexity ↓ | | LLAMA-3.1 | GEMMA-2 | YI-1.5 | DEEPSEEK | QWEN2.5 |
|---|---|---|---|---|---|---|
| **Method** | Sparsity | 8B | 9B | 9B | 7B | 7B |
| Wanda | 60% | 21.94 | 16.74 | 11.40 | 11.41 | 13.75 |
| + DSnoT | 60% | 21.94 | 16.69 | 11.38 | 11.40 | 13.75 |
| **+ SparseSwaps** | 60% | **19.75** | **16.01** | **10.07** | **10.93** | **13.16** |
| RIA | 60% | 19.73 | 16.19 | 10.73 | 11.80 | 12.63 |
| + DSnoT | 60% | 19.73 | 16.22 | 10.73 | 11.80 | 12.63 |
| **+ SparseSwaps** | 60% | **18.47** | **15.44** | **9.98** | **10.79** | **12.47** |
| Wanda | 2:4 | 24.82 | 17.45 | 11.76 | 11.77 | 14.53 |
| + DSnoT | 2:4 | 22.79 | 16.79 | 10.84 | 11.70 | 14.40 |
| **+ SparseSwaps** | 2:4 | **20.17** | **16.30** | **10.73** | **11.70** | **13.95** |
| RIA | 2:4 | 23.96 | 16.88 | 11.29 | 12.03 | 13.58 |
| + DSnoT | 2:4 | 24.26 | 16.82 | 10.57 | 12.03 | 13.85 |
| **+ SparseSwaps** | 2:4 | **20.90** | **16.33** | **10.50** | **11.80** | **13.28** |
| **Accuracy ↑** | | LLAMA-3.1 | GEMMA-2 | YI-1.5 | DEEPSEEK | QWEN2.5 |
| **Method** | Sparsity | 8B | 9B | 9B | 7B | 7B |
| Wanda | 60% | 48.18% | 63.39% | 53.59% | 50.74% | 59.26% |
| + DSnoT | 60% | 48.18% | 63.49% | 53.79% | 50.75% | 59.26% |
| **+ SparseSwaps** | 60% | **50.78%** | **63.84%** | **54.84%** | **51.02%** | **60.15%** |
| RIA | 60% | 49.56% | 64.37% | 52.81% | 50.92% | 59.84% |
| + DSnoT | 60% | 49.56% | **64.43%** | 52.96% | 50.83% | 59.81% |
| **+ SparseSwaps** | 60% | **51.02%** | 64.32% | **54.45%** | **51.47%** | **61.22%** |
| Wanda | 2:4 | 46.80% | 63.73% | **52.58%** | **51.02%** | 59.52% |
| + DSnoT | 2:4 | 47.01% | 63.66% | 52.16% | 50.78% | 59.09% |
| **+ SparseSwaps** | 2:4 | **48.83%** | **64.70%** | 52.43% | 50.36% | **59.92%** |
| RIA | 2:4 | 47.87% | 63.87% | **52.68%** | 51.22% | 58.66% |
| + DSnoT | 2:4 | 47.13% | 64.17% | 51.36% | 49.86% | 59.72% |
| **+ SparseSwaps** | 2:4 | **49.90%** | **64.60%** | 52.30% | **51.46%** | **60.31%** |

justified when amortized over many LLM inference requests, we note that the overhead grows only linearly with the number of 1-swap iterations $T_{\max}$. Table 1 shows that few iterations already yield substantial gains in both perplexity and local error reduction, especially at higher sparsity.

Table 3 reports wall-clock times for pruning LLAMA-3.1-8B to 60% sparsity on a single H100 GPU. The $T_{\max} = 0$ baseline includes calibration data sampling, Wanda pruning, Gram matrix computation, and evaluation; each additional iteration of SparseSwaps adds a relatively small overhead. For comparison, Wanda and SparseGPT take approximately 4 and 10 minutes, respectively. We note that our implementation can be further optimized and that the algorithm is fully parallelizable across rows.

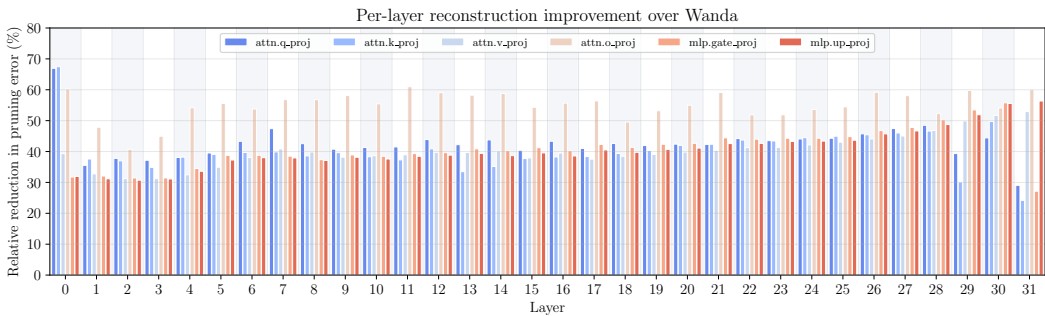

Figure 1: Per-layer relative reduction in local pruning error compared to Wanda. The plot shows result for LLAMA-3.1-8B, 60% unstructured sparsity and 100 1-swap iterations.

Table 3: Wall-clock time for applying SparseSwaps to LLAMA-3.1-8B at 60% sparsity on a single H100 GPU.

| $T_{\max}$ | 0 | 1 | 2 | 5 | 10 | 25 |
|---|---|---|---|---|---|---|
| Wall-clock time | 8m15s | 10m17s | 12m7s | 17m20s | 26m13s | 52m29s |

**Effect of the number of reconstruction samples.** Figure 2 in the appendix shows the perplexity versus the number of reconstruction samples for 50% and 60% unstructured sparsity when using Wanda as well as SparseSwaps with a Wanda warmstart. We observe that the perplexity decreases drastically when using more samples, which leads to SparseSwaps slightly outperforming Wanda for 50% sparsity, despite its advantage typically being larger at higher sparsity. We emphasize that the number of reconstruction samples does not affect SparseSwaps's swap evaluation efficiency: the Gram matrix $G = XX^\top$ has fixed size $d_{in} \times d_{in}$ regardless of $B$.

## 4 CONCLUSION

We revisited the mask selection problem for post-training pruning and showed that it can be made substantially more tractable, even at LLM scale. We observed that row decoupling via equal per-row sparsity yields independent subproblems, and that individual 1-swaps can be evaluated in $\mathcal{O}(1)$ time using the Gram matrix $G = XX^\top$. This enables tractable optimization of the true row-wise quadratic loss on GPUs. The resulting method, SparseSwaps, is warm-start agnostic, nearly hyperparameter-free, and scalable. It consistently reduces per-layer pruning error and improves perplexity and zero-shot accuracy across modern GPT architectures.

Our work is not without limitations. While per-row sparsity is not necessarily detrimental for LLMs, our approach is restricted to that setting and only partially adapts to truly unstructured sparsity; in its current form, the algorithm can handle unstructured sparsity but cannot reallocate sparsity levels across rows. Furthermore, runtime and memory remain non-trivial for large architectures.

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

# A APPENDIX

## A.1 USE OF LARGE LANGUAGE MODELS

Large language models were used to aid in writing (polishing text) as well as to help with the implementation of code components, including both the methods and the generation of plots. They also served as a tool for brainstorming research ideas and refining development approaches to address the challenges explored in this paper.

## A.2 FURTHER RESULTS

Table 4: Perplexity (↓, lower is better) comparison on WikiText. We report SparseSwaps refinement with magnitude warmstart for 50% and 60% sparsity. Best values are highlighted in **bold**. We omit standard deviations for legibility.

| **Perplexity ↓** | | **LLAMA-3.1** | **GEMMA-2** | **DEEPSEEK** |
|---|---|---|---|---|
| **Method** | Sparsity | 8B | 9B | 7B |
| Magnitude | 50% | 68.89 | 31.87 | 25.05 |
| **+ SparseSwaps** | 50% | 52.26 | 19.11 | 16.23 |
| Magnitude | 60% | 3486.26 | 184.52 | 330.07 |
| **+ SparseSwaps** | 60% | 264.92 | 60.04 | 80.24 |

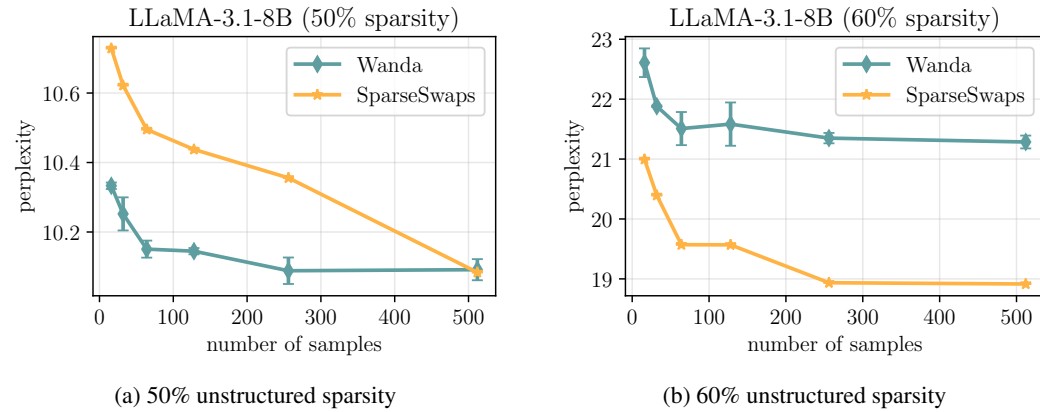

(a) 50% unstructured sparsity                    (b) 60% unstructured sparsity

Figure 2: Perplexity versus the number of reconstruction samples for unstructured sparsity using Wanda warmstart.

