# OpenReview forum: "SparseSwaps: Tractable LLM Pruning Mask Refinement at Scale"
_ICLR.cc/2026/Conference — Submitted to ICLR 2026_

### Official Review · Reviewer_1jXE · 2025-10-26

**Soundness:** 3
**Presentation:** 3
**Contribution:** 1
**Rating:** 2
**Confidence:** 4

**Summary:**

The paper proposes a heuristic local optimization algorithm for finding better pruning masks for LLMs.
Achieves better results than DSnoT.

**Strengths:**

The proposed algorithm is sound and correct.

**Weaknesses:**

- "EQUAL SPARSITY-LEVEL ACROSS ROWS MUST NOT BE DETRIMENTAL" is supported by the findings in Wanda, but it is not supported in SparseGPT and ADMM pruning [1]
- SVD trick on calibration data might be unnecessary, since the reconstruction error can also be written as $tr((W_p - W)(X^TX)(W_p - W)^T)$ and $X^TX$ has $d \times d$ shape. Or in other words, why do costly SVD, when one can just store Hessian ($X^TX$) for the layer-wise reconstruction problem?
- There is no explicit runtime mentioned, only something vague in the last sentence.
- Why would I use the SparseSwaps algorithm over SparseGPT/ADMM? For example, SparseSwaps achieves 19.75 perplexity on 60% Llama-3.1-8B, while ADMM achieves 13.92. And ADMM/SparseGPT are fast.
- Also, something weird is happening for 50% sparsity (Table 2), where SparseSwaps did not provide any benefit over Wanda.
- "Optional Weight Reconstruction" section does not make much sense, since computing $(X_uX_u^T)^-1$ for each row would need way too many matrix inverses. Approaches such as [1] are much better.
- Metrics for original dense models should also be presented (e.g., in Table 1)

[1] Boža, Vladimír. "Fast and Effective Weight Update for Pruned Large Language Models." Transactions on Machine Learning Research.

**Questions:**

See weaknesses.

---

> ### Author Response · Authors · 2025-11-21
>
> Thanks for taking the time to evaluate and help improving our manuscript. We greatly appreciate the constructive feedback.
>
> > "EQUAL SPARSITY-LEVEL ACROSS ROWS MUST NOT BE DETRIMENTAL" is supported by the findings in Wanda, but it is not supported in SparseGPT and ADMM pruning
>
> We think there might be a misunderstand here due to poor communication from our side; what we meant is that equal per-row sparsity **is not necessarily** detrimental (we will adapt the phrasing). Our intended claim is that there is indication in the LLM pruning literature that fixed row-wise sparsity can work better or as good as unstructured sparsity (e.g., Sun et al. demonstrate this for Wanda and Magnitude pruning). Furthermore, our results demonstrate that solving the objective row wise nonetheless leads to large reductions in local loss. For N:M sparsity, which is (at least currently) more practically relevant for GPU acceleration, the sparsity level is equal over all rows by definition.
>
> Regarding the findings of SparseGPT and ADMM: mathematically, unstructured sparsity constraints obviously give a higher level of flexibility than per-row sparsity constraints. We do not claim otherwise. We simply state that per-row sparsity constraints, in the setting of LLMs specifically, are not necessarily detrimental to performance, as the results by Sun et al. show. Even in ADMM pruning, the authors state that the per-row sparsity constraint suggested by Sun et al. is only "slightly detrimental". We will make sure to change our phrasing to a more precise formulation.
>
>
> > SVD trick on calibration data might be unnecessary, since the reconstruction error can also be written [...]
>
> We agree that the objective can be equivalently formulated only be the Gram Matrix, which would then in turn do not require any more computation of the SVD. In practice, leaving out the SVD does not make that much of a big difference, since the SVD comparably takes up only a relatively small fraction of the runtime (cf. the block timing summary in the answer below). However, our current (admittedly suboptimal) implementation relies on rather expensive data storing and loading mechanisms to have the necessary data available when pruning the individual matrices, while only a single forward pass for the entire transformer block is required (as is equally done in the code of SparseGPT, Wanda, ADMM pruning). This (again, admittedly suboptimal but working) implementation can be sped up if we follow the approach of e.g. SparseGPT and directly collect the Gram matrix as a buffer during the forward pass.
>
> We would like to emphasize that this is an equivalent formulation, the results are not affected by this change. But of course we will change parts of our phrasing in the upcoming revision to make clear that the SVD is not necessarily needed, albeit one approach to reducing the complexity of the problem.

---

> > ### Author Response · Authors · 2025-11-21
> >
> > > There is no explicit runtime mentioned, only something vague in the last sentence.
> >
> > Thank you for the suggestion. By definition, mask refinement methods like SparseSwaps or DsNOT add computational overhead compared to the baseline methods they refine. That being said, SparseSwaps is a resource-intensive method (at least w.r.t. compute and our current implementation). We argue in the paper that the additional compute is worth it because it can be amortized over many inference requests, given the increased inference demands of LLMs. Furthermore, the additional compute is only linear in the number of 1-swap iterations $T_{\max}$, which is typically set to a small value. To address your concern, we have added the runtimes of LLaMA-3.1-8B on a single H100 GPU in the exact same setup as in Table 2 in the paper below (60% Sparsity), and will include a detailed discussion and analysis in the upcoming revision. Note that the case of 0 iterations corresponds to the full runtime of the method without doing any swaps, i.e., the sampling of calibration data, layerwise pruning using Wanda, including per-layer data collection and SVD computation for SparseSwaps, as well as the final Wikitext and EleutherAI evaluation. For reference, we have also computed the runtime of SparseGPT, which amounts to 35 minutes.
> >
> >
> > | $T_{\text{max}}$ | 0 | 1 | 2 | 5 | 10 | 25 | 50 | 100 | 200 |
> > | --- | --- | --- | --- | --- | --- | --- | --- | --- | --- |
> > | **Walltime (hours)** | 2h45m | 2h54m | 3h13m | 3h18m | 3h35m | 4h01m | 4h51m | 6h22m | 10h03m |
> >
> >
> > Some remarks:
> > - We think that our implementation can be further improved w.r.t. efficiency (as we have also argued above). As the comparison between $T_{\max} = 0$ and $T_{\max} > 0$ as well as the per-transformer-block breakdown of a LLAMA-3.1-8B below (25 iterations, 50% sparsity) highlight, our implementation currently spends roughly half of the runtime on data operations; in part, this is due to an implementation where we do a single forward pass through each transformer block, save the necessary datafiles for each single layer matrix to the hard drive, and then load them back into memory when running the SparseSwaps algorithm for the layer. This is certainly a suboptimal implementation which we will improve for the camera-ready version and publication of the code.
> > - Nevertheless, we acknowledge that SparseSwaps incurs non-trivial computational overhead. However, we emphasize that the algorithm is fully parallelizable across rows; one could solve each individual row on its own GPU if necessary.
> > - Our main goal was to show that an greedy-exact approach like SparseSwaps is realizable; the efficiency can be drastically improved by relaxing the exactness requirement, and to allow e.g. doing more than one swap per iteration, and then only updating the cost vectors.
> >
> > ```
> > ============================================================
> > BLOCK 0 TIMING SUMMARY
> > ============================================================
> > Data Collection:     88.485s (31.4%)
> > Matrix Processing:   193.336s (68.6%)
> >   - Data loading   : 42.357s (15.0%)
> >   - SVD Compression: 5.099s (1.8%)
> >   - SparseSwaps   : 142.532s (50.6%)
> >   - Error measurement: 0.079s (0.0%)
> >   - Cleanup        : 3.269s (1.2%)
> > BLOCK TOTAL:         281.821s (100.0%)
> > ============================================================
> > ```

---

> > > ### Author Response · Authors · 2025-11-21
> > >
> > > > Why would I use the SparseSwaps algorithm over SparseGPT/ADMM? For example, SparseSwaps achieves 19.75 perplexity on 60% Llama-3.1-8B, while ADMM achieves 13.92. And ADMM/SparseGPT are fast.
> > >
> > > Let us clarify that the focus of our work was to demonstrate that exact mask refinement is realizable at LLM scale and can improve existing pruning methods such as Wanda. Our objective is to refine pruning mask posthoc, as e.g., DsNOT does. There are of course methods such as SparseGPT, which solve the mask selection and weight reconstruction steps jointly. As such, we do not claim to beat all other sparsification methods, and we do think that the work presented in this paper is both complementary to existing approaches and a relevant contribution to the research literature.
> > >
> > > That being said, both SparseGPT and ADMM could be combined with SparseSwaps to improve the intermediate pruning steps. In particular, note that already a couple of swaps can lead to significant reductions in the local pruning error.
> > >
> > > **SparseGPT:**
> > > SparseGPT is based on the OBS criterion, which ensures that the pruning and reconstruction steps are jointly optimal for pruning *one weight at a time*. SparseGPT trades off speed for accuracy to ensure scalability to large models by pruning small blocks of weights at a time instead of single weights as suggested by the theoretical analysis. Since these pruning are not necessarily greedy-optimal, SparseSwaps could be used to refine the pruning step.
> > >
> > > **ADMM-based methods:**
> > > While the ADMM algorithm itself provably converges the pruning step in the proposed ADMM-based method itself is heuristic. Hence SparseSwaps could be used to refine these intermediate pruning steps.
> > >
> > >
> > > > Also, something weird is happening for 50% sparsity (Table 2), where SparseSwaps did not provide any benefit over Wanda.
> > >
> > > We should have discussed that more explicitly in the paper (and we will). Overall, we found that SparseSwaps works best when the warmstart is of suboptimal quality. That includes both weak warmstart masks or relatively high, impactful sparsity levels or patterns. If the warmstart mask already results in a reasonably good solution, then SparseSwaps can still significantly improve upon the local solution, albeit with little to no improvements in overall perplexity (and sometimes even slightly worsening it).
> > >
> > > We attribute this to two points: a) the local error is a mere approximation to the overall perplexity/loss, and b) just optimizing the mask itself is limited in preserving outputs of the overall networks. Especially the latter might be mitigatable when allowing further reconstruction of the remaining, non-pruned weights. Only choosing the pruning mask for individual layers yields an approximation that has much less functional expressivity than choosing the pruning mask and adapting the remaining weights jointly.
> > >
> > > > "Optional Weight Reconstruction" section does not make much sense
> > >
> > > We agree that solving the reconstruction problem using matrix inversions is costly. This short paragraph was meant to highlight, that one can potentially do weight reconstruction after running SparseSwaps. We did not intend to overemphasize this, in particular, we do not reconstruct the weights in any part of our algorithm; the focus of our work lied completely on the optimization of the mask. We will remove the corresponding sentences from the upcoming revision.
> > >
> > >
> > > > Metrics for original dense models should also be presented (e.g., in Table 1)
> > >
> > > Fair point, we will add the dense baseline metrics to the relevant tables. Thanks!
> > >
> > > Thank you again for your review, please let us know if further clarification is needed.

---

> > > > ### Comment · Reviewer_1jXE · 2025-11-26
> > > >
> > > > My questions have been answered.
> > > > While I think that this is solid work, I think that its impact is not good enough for a venue like ICLR.
> > > >
> > > > I will maintain my score.

---

### Official Review · Reviewer_Q6G3 · 2025-10-27

**Soundness:** 3
**Presentation:** 3
**Contribution:** 3
**Rating:** 8
**Confidence:** 4

**Summary:**

This paper studies layer-wise pruning for LLMs by reframing the mask-selection objective as a GPU-friendly local optimization problem that monotonically reduces the reconstruction loss. It is argued that exactly solving the combinatorial mask selection is intractable, hence existing methods must rely on surrogates that ignore within-row interactions. The authors propose to make the per-row objective separable by enforcing equal sparsity per row or N:M blocks, compress the calibration activations via an SVD-based unitary transformation to shrink the data dimension, and then apply 1-swap evaluations with efficient incremental updates to greedily pick the best kept or pruned exchange per row. The proposed method caches the weighted contributions and maintains a running sum of pruned rows; each candidate 1-swap is scored via a precomputed norm and a dot-product with the current residual, which gives fast monotone improvements under per-row or N:M constraints. For implementation, the algorithm warm-starts from any mask, performs up to certain swap iterations per row, and optionally applies a least-squares weight reconstruction on the kept indices. Experiments show that the proposed method yields up to 70% reductions in per-layer pruning error over previous method and attain consistent performance gains at higher sparsity. Improvements at milder sparsity can be lower.

**Strengths:**

The paper is well-written.  The main observations on row separability, unitary invariance and SVD compression, and exact 1-swap with incremental updates are convincing and directly related to the complexity bottlenecks of pruning LLMs. Using exact 1-swap search over the true objective is a new and interesting approach compared to previous LLM pruning methods which often optimize surrogates.  The proposed method is well-motived based on the observations, with detailed discussion on complexity and memory trade-offs. Experiments on several LLMs show good error reduction and performance gains. Discussion is also given for the cases where local loss reductions don’t translate to performance gains.

**Weaknesses:**

I don't see any major weaknesses. Perhaps the authors should consider taking account of structures within q/k/v or MLP sub-blocks into their approach to understand why some layers benefit more than others.

**Questions:**

1. It might be helpful to see peak per-layer GPU memory, wall-clock and GPU cost for different T_max's and sparsities.
2. Are perplexity and zero-shot accuracy sensitive to calibration corpus domain shifts and the number of calibration tokens?

---

> ### Author Response · Authors · 2025-11-21
>
> We thank you for your positive review and for recognizing the strengths of our work.
>
> > Perhaps the authors should consider taking account of structures within q/k/v or MLP sub-blocks into their approach to understand why some layers benefit more than others.
>
> If we understand you correctly, you suggest to swap-optimize a function that contains more than one matrix and potentially non-linear operations in between? The structure that made our approach feasible is tied to the decoupling of rows and the linearity of matrix multiplication. Exactly solving the problem for multiple matrices at a time is significantly more complex, in particular we lose the structure that made explicit computation of the local loss change due to 1-swaps tractable. We are currently investigating whether we can derive exact steps for parts of a Transformer block and will get back to you if successful.
>
>
>
> > Are perplexity and zero-shot accuracy sensitive to calibration corpus domain shifts and the number of calibration tokens?
>
> As the warmstart masks also depend on that choice, we have not evaluated the method under different sequence lengths and on other datasets. We think that if the calibration dataset contains enough samples of sufficient length (implying that the warmstart masks are of somewhat reasonable quality), SparseSwaps will equally work. Note that we have included experiments with different number of calibration samples in Figure 2 in the appendix (discussed in the main part starting from line 450). We further ran an ablation study comparing Wanda and SparseSwaps when varying the sequence length while keeping the number of calibration samples fixed at 128. Both methods were equally impacted by the sequence length, which overall was rather minor in terms of final perplexity. We will add a detailed discussion of this to the camera-ready revision.

---

> > ### Author Response · Authors · 2025-11-21
> >
> > > It might be helpful to see peak per-layer GPU memory, wall-clock and GPU cost for different T_max's and sparsities.
> >
> > Thank you for the suggestion. By definition, mask refinement methods like SparseSwaps or DsNOT add computational overhead compared to the baseline methods they refine. That being said, SparseSwaps is a resource-intensive method (at least w.r.t. compute and our current implementation). We argue in the paper that the additional compute is worth it because it can be amortized over many inference requests, given the increased inference demands of LLMs. Furthermore, the additional compute is only linear in the number of 1-swap iterations $T_{\max}$, which is typically set to a small value. To address your concern, we have added the runtimes of LLaMA-3.1-8B on a single H100 GPU in the exact same setup as in Table 2 in the paper below (60% Sparsity), and will include a detailed discussion and analysis in the upcoming revision. Note that the case of 0 iterations corresponds to the full runtime of the method without doing any swaps, i.e., the sampling of calibration data, layerwise pruning using Wanda, including per-layer data collection and SVD computation for SparseSwaps, as well as the final Wikitext and EleutherAI evaluation. For reference, we have also computed the runtime of SparseGPT, which amounts to 35 minutes.
> >
> >
> > | $T_{\text{max}}$ | 0 | 1 | 2 | 5 | 10 | 25 | 50 | 100 | 200 |
> > | --- | --- | --- | --- | --- | --- | --- | --- | --- | --- |
> > | **Walltime (hours)** | 2h45m | 2h54m | 3h13m | 3h18m | 3h35m | 4h01m | 4h51m | 6h22m | 10h03m |
> >
> >
> > Some remarks:
> > - We think that our implementation can be further improved w.r.t. efficiency. As the comparison between $T_{\max} = 0$ and $T_{\max} > 0$ as well as the per-transformer-block breakdown of a LLAMA-3.1-8B below (25 iterations, 50% sparsity) highlight, our implementation currently spends roughly half of the runtime on data operations; in part, this is due to an implementation where we do a single forward pass through each transformer block, save the necessary datafiles for each single layer matrix to the hard drive, and then load them back into memory when running the SparseSwaps algorithm for the layer. This is certainly a suboptimal implementation which we will improve for the camera-ready version and publication of the code.
> > - Nevertheless, we acknowledge that SparseSwaps incurs non-trivial computational overhead. However, we emphasize that the algorithm is fully parallelizable across rows; one could solve each individual row on its own GPU if necessary.
> > - Our main goal was to show that an greedy-exact approach like SparseSwaps is realizable; the efficiency can be drastically improved by relaxing the exactness requirement, and to allow e.g. doing more than one swap per iteration, and then only updating the cost vectors.
> >
> > ```
> > ============================================================
> > BLOCK 0 TIMING SUMMARY
> > ============================================================
> > Data Collection:     88.485s (31.4%)
> > Matrix Processing:   193.336s (68.6%)
> >   - Data loading   : 42.357s (15.0%)
> >   - SVD Compression: 5.099s (1.8%)
> >   - SparseSwaps   : 142.532s (50.6%)
> >   - Error measurement: 0.079s (0.0%)
> >   - Cleanup        : 3.269s (1.2%)
> > BLOCK TOTAL:         281.821s (100.0%)
> > ============================================================
> > ```
> >
> > Memorywise, when processing each layer, we only need to keep the (d_in x d_in) matrix $S$ in memory. When performing the SVD, we never realize the full (d_in x B) matrix, but rather perform batched computation of $XX^T$ and compute the SVD by diagonalization of the resulting (d_in x d_in) matrix. In the individual iterations over the rows, we mainly need to keep the (d_in x d_in) matrix $S$ in memory, as well as the (d_in x 1) vector $s$ for the current cost vector. The practical implementation is a bit more convoluted and keeps additional buffers. We conducted some preliminary additional experiments to measure the peak memory requirements when pruning to 50\% sparsity for a LLAMA-3.1-8B on a single H100 GPU, and got the following results:
> >
> > - Wanda: 25590 MiB
> > - SparseSwaps with Wanda warmstart: 49478 MiB
> >
> > Please let us know if further clarification is needed.

---

### Official Review · Reviewer_ohzU · 2025-10-29

**Soundness:** 3
**Presentation:** 3
**Contribution:** 3
**Rating:** 6
**Confidence:** 2

**Summary:**

This paper addresses the problem of finding optimal pruning masks for Large Language Models (LLMs) in a post-training, retraining-free setting. The authors correctly identify that state-of-the-art methods solve a layer-wise reconstruction error minimization problem, but that solving this problem exactly is computationally intractable due to both the combinatorial search space and, more critically, the prohibitive memory cost of caching intermediate values required for the optimization.

1. The paper introduces SparseSwaps, a method to refine an existing pruning mask by making this optimization problem tractable. The core of the work rests on three key insights:
2. Row-wise Decoupling: Enforcing equal sparsity per row (a common practice in LLM pruning) decouples the optimization problem, allowing each row of the weight matrix to be handled independently.
3. SVD-based Compression: Leveraging the unitary invariance of the Frobenius norm, the high-dimensional calibration data matrix X can be compressed via SVD into a much smaller matrix X' without changing the optimization objective. This elegantly solves the memory bottleneck.

Efficient 1-Swap Local Search: The authors propose an iterative local search algorithm that efficiently evaluates all possible "1-swaps" (exchanging one pruned weight for one unpruned weight) by pre-computing intermediate values and using incremental updates. This allows for monotonic improvement of the true row-wise reconstruction error.
The proposed SparseSwaps algorithm is presented as a post-hoc refinement step that can be applied to row-wise sparse masks (e.g., from Wanda or RIA). Extensive experiments on a suite of modern LLMs (Llama-3.1, Gemma-2, etc.) demonstrate that SparseSwaps consistently improves perplexity and zero-shot accuracy over strong baselines for both unstructured and semi-structured (2:4) sparsity.

**Strengths:**

1.The paper correctly identifies a major practical limitation of sota layer-wise LLM pruning methods: the computational intractability.
2.This paper proposes  three clever insights includes Row decouping, SVD Compressing and 1-Swap optimization that significantly reduce the problem's complexity with clear mathematics analysis.
3.The paper provides compelling evidence for the effectiveness of SparseSwaps across multiple modern LLM architectures and sparsity patterns (unstructured, 2:4 N:M).

**Weaknesses:**

1. Constraint to Per-Row Sparsity: The first insight, which enables the method's tractability, is also its main limitation. By decoupling the rows, the algorithm cannot reallocate sparsity between different rows of a weight matrix. This restricts its ability to find a truly optimal unstructured mask at the layer level, as the sparsity budget for each row is fixed by the warm-start mask. The authors acknowledge this limitation in the conclusion.
2. Computational Overhead: While the paper argues the cost is amortizable, SparseSwaps is inherently more computationally expensive than the one-shot methods it refines. A more detailed analysis of the practical wall-clock time and peak memory usage on standard hardware (e.g., for a 7B model on an A100) would be beneficial for practitioners to gauge the trade-off between performance gain and computational cost. The theoretical complexity is given, but its real-world implication remains somewhat abstract.
3. Lacks ablation on the key findings and design choices i.e. p-u interaction and Tmax.

**Questions:**

1. Practical Cost Analysis: Could you provide concrete wall-clock timings for running SparseSwaps (e.g., for T_max=100 iterations) on a model like Llama-3.1-8B and compare it to the runtime of the baseline methods it refines (Wanda/RIA)? This would provide a clearer picture of the practical cost involved.
2. Overfitting and Regularization: Your analysis showing that minimizing local error can sometimes hurt perplexity (Table 2, 50% sparsity) is very interesting. Have you considered any mechanisms to mitigate this? For example, could one use a small validation set of calibration data to implement early stopping for the swap iterations on a per-layer basis?
3. Exploring Inter-Row Swaps: Given the limitation of the per-row constraint, have you considered a hybrid approach? For instance, after the per-row optimization converges, one could perform a limited number of swaps between rows (e.g., swapping a pruned weight in a "low-impact" row for a kept weight in a "high-impact" row). Do you believe such an extension would be feasible and/or beneficial?

---

> ### Author Response · Authors · 2025-11-21
>
> We thank you for your review and constructive feedback. We address your concerns below:
>
>
> ### Weaknesses and Questions
>
> > 1. Constraint to Per-Row Sparsity: The first insight, which enables the method's tractability, is also its main limitation. By decoupling the rows, the algorithm cannot reallocate sparsity between different rows of a weight matrix. This restricts its ability to find a truly optimal unstructured mask at the layer level, as the sparsity budget for each row is fixed by the warm-start mask. The authors acknowledge this limitation in the conclusion. [...] Exploring Inter-Row Swaps: Given the limitation of the per-row constraint, have you considered a hybrid approach? For instance, after the per-row optimization converges, one could perform a limited number of swaps between rows (e.g., swapping a pruned weight in a "low-impact" row for a kept weight in a "high-impact" row). Do you believe such an extension would be feasible and/or beneficial?
>
> In its current form, our algorithm is indeed limited to sparsity types which ensure a fixed level of sparsity per row. We highlight once again that we argue a) that this is not a problem for modern sparsity patterns like N:M sparsity, which inherit this property and the rows are by definition decoupled, and b) that we think that enforcing per-row sparsity instead of unstructured sparsity must not necessarily be detrimental, as highlighted by Sun et al. (2023). In their mathematical motivation, also methods like SparseGPT are bound to per-row sparsity; we think that a similar approach could be taken to adapt SparseSwaps to unstructured sparsity patterns. Your suggestion to explore inter-row swaps is interesting, thanks! We think that this could yield some benefits. This will take some additional effort to implement, but we will discuss your suggestion in the camera-ready revision.
>
> > 2. Computational Overhead: While the paper argues the cost is amortizable, SparseSwaps is inherently more computationally expensive than the one-shot methods it refines. A more detailed analysis of the practical wall-clock time and peak memory usage on standard hardware (e.g., for a 7B model on an A100) would be beneficial for practitioners to gauge the trade-off between performance gain and computational cost.
>
> Thank you for the suggestion. By definition, mask refinement methods like SparseSwaps or DsNOT add computational overhead compared to the baseline methods they refine. That being said, SparseSwaps is a resource-intensive method (at least w.r.t. compute and our current implementation). We argue in the paper that the additional compute is worth it because it can be amortized over many inference requests, given the increased inference demands of LLMs. Furthermore, the additional compute is only linear in the number of 1-swap iterations $T_{\max}$, which is typically set to a small value. To address your concern, we have added the runtimes of LLaMA-3.1-8B on a single H100 GPU in the exact same setup as in Table 2 in the paper below (60% Sparsity), and will include a detailed discussion and analysis in the upcoming revision. Note that the case of 0 iterations corresponds to the full runtime of the method without doing any swaps, i.e., the sampling of calibration data, layerwise pruning using Wanda, including per-layer data collection and SVD computation for SparseSwaps, as well as the final Wikitext and EleutherAI evaluation. For reference, we have also computed the runtime of SparseGPT, which amounts to 35 minutes.
>
>
> | $T_{\text{max}}$ | 0 | 1 | 2 | 5 | 10 | 25 | 50 | 100 | 200 |
> | --- | --- | --- | --- | --- | --- | --- | --- | --- | --- |
> | **Walltime (hours)** | 2h45m | 2h54m | 3h13m | 3h18m | 3h35m | 4h01m | 4h51m | 6h22m | 10h03m |
>
>
> Some remarks:
> - We think that our implementation can be further improved w.r.t. efficiency. As the comparison between $T_{\max} = 0$ and $T_{\max} > 0$ as well as the per-transformer-block breakdown of a LLAMA-3.1-8B below (25 iterations, 50% sparsity) highlight, our implementation currently spends roughly half of the runtime on data operations; in part, this is due to an implementation where we do a single forward pass through each transformer block, save the necessary datafiles for each single layer matrix to the hard drive, and then load them back into memory when running the SparseSwaps algorithm for the layer. This is certainly a suboptimal implementation which we will improve for the camera-ready version and publication of the code.
> - Nevertheless, we acknowledge that SparseSwaps incurs non-trivial computational overhead. However, we emphasize that the algorithm is fully parallelizable across rows; one could solve each individual row on its own GPU if necessary.
> - Our main goal was to show that an greedy-exact approach like SparseSwaps is realizable; the efficiency can be drastically improved by relaxing the exactness requirement, and to allow e.g. doing more than one swap per iteration, and then only updating the cost vectors.

---

> > ### Author Response · Authors · 2025-11-21
> >
> > ```
> > ============================================================
> > BLOCK 0 TIMING SUMMARY
> > ============================================================
> > Data Collection:     88.485s (31.4%)
> > Matrix Processing:   193.336s (68.6%)
> >   - Data loading   : 42.357s (15.0%)
> >   - SVD Compression: 5.099s (1.8%)
> >   - SparseSwaps   : 142.532s (50.6%)
> >   - Error measurement: 0.079s (0.0%)
> >   - Cleanup        : 3.269s (1.2%)
> > BLOCK TOTAL:         281.821s (100.0%)
> > ============================================================
> > ```
> >
> > Memorywise, when processing each layer, we only need to keep the (d_in x d_in) matrix $S$ in memory. When performing the SVD, we never realize the full (d_in x B) matrix, but rather perform batched computation of $XX^T$ and compute the SVD by diagonalization of the resulting (d_in x d_in) matrix. In the individual iterations over the rows, we mainly need to keep the (d_in x d_in) matrix $S$ in memory, as well as the (d_in x 1) vector $s$ for the current cost vector. The practical implementation is a bit more convoluted and keeps additional buffers. We conducted some preliminary additional experiments to measure the peak memory requirements when pruning to 50\% sparsity for a LLAMA-3.1-8B on a single H100 GPU, and got the following results:
> >
> > - Wanda: 25590 MiB
> > - SparseSwaps with Wanda warmstart: 49478 MiB
> >
> >
> > > 3. Lacks ablation on the key findings and design choices i.e. p-u interaction and Tmax.
> >
> > We provided an ablation study on $T_{\max}$ in Table 2 in the main part. Could you elaborate what ablation you would like to see w.r.t. the p-u interaction? We are not entirely sure what you mean here, but are happy to provide any ablation study we can provide given the rebuttal timeframe.
> >
> >
> > > Overfitting and Regularization: Your analysis showing that minimizing local error can sometimes hurt perplexity (Table 2, 50% sparsity) is very interesting. Have you considered any mechanisms to mitigate this? For example, could one use a small validation set of calibration data to implement early stopping for the swap iterations on a per-layer basis?
> >
> >
> > We agree that this is an interesting observation and we should have made our findings in that regard more explicit in the paper (and we will). Overall, we found that SparseSwaps works best when the warmstart is of suboptimal quality. That includes both weak warmstart masks or relatively high, impactful sparsity levels or patterns. If the warmstart mask already results in a reasonably good solution, then SparseSwaps can still significantly improve upon the local solution, albeit with little to no improvements in overall perplexity (and sometimes even slightly worsening it).
> >
> > We attribute this to two points: a) the local error is a mere approximation to the overall perplexity/loss, and b) just optimizing the mask itself is limited in preserving outputs of the overall networks. Especially the latter might be mitigatable when allowing further reconstruction of the remaining, non-pruned weights. Only choosing the pruning mask for individual layers yields an approximation that has much less functional expressivity than choosing the pruning mask and adapting the remaining weights jointly.
> >
> > Thank you for the suggestion of using a small validation dataset. While we think that this does not address the main issue, we will implement it and get back to you with the results.
> >
> > Please let us know if further clarification is needed.

---

### Official Review · Reviewer_svrG · 2025-10-29

**Soundness:** 2
**Presentation:** 2
**Contribution:** 2
**Rating:** 4
**Confidence:** 3

**Summary:**

This paper introduces SparseSwaps, a scalable and tractable post-training mask refinement algorithm for pruning LLMs. Empirical validation is performed across various large-scale open-source LLM, reporting significant improvements over existing pruning methods in terms of per-layer pruning error, perplexity, and zero-shot accuracy.

**Strengths:**

1. To address the core bottlenecks in LLM pruning, this paper propose an integrated framework that combines row decoupling, SVD-based compression, and a 1-swap strategy. This approach achieves substantial improvements over existing pruning methods such as DSnoT and Wanda.
2. This paper is well-motivated by three insights in Sec. 2.
3. The experiments report consistent and sometimes substantial improvements on both local pruning loss and downstream task metrics across multiple model families.
4. The theoretical analysis in this paper is convincing, as it not only explains the effectiveness of SparseSwaps but also clarifies why previous methods are suboptimal.

**Weaknesses:**

1. The experiments in this paper are somewhat limited. Although results are provided for five LLM models, all of them are language models. It remains unclear how SparseSwaps performs on vision models or other types of Transformer architectures. This limitation constrains the generality and comprehensiveness of the evaluation.

2. The paper does not provide the runtime of SparseSwaps on different models or comparisons with baselines, which makes it difficult to evaluate the proposed method.

3. The paper lacks a theoretical or experimental characterization of the convergence behavior of 1-swap.

4. The experiments in the paper appear to use a fixed setup (see line 350), lacking evaluations of the method under different sequence lengths and on other datasets.

**Questions:**

1. Can the authors provide a comparison of runtime and memory usage between SparseSwaps and other baselines, using the same experimental setup?
2. Is the effectiveness of SparseSwaps limited to specific data distributions and experimental setups? Could the authors provide additional experimental results under different calibration data settings?
3. Since SparseSwaps depends on warm-start masks (such as outputs from Wanda or RIA), how sensitive is the method to the quality of these masks? If the warm-start mask is of low quality (e.g., randomly generated masks or poor heuristic pruning masks), can SparseSwaps still effectively minimize pruning errors?
4. Are there any plans to expand the theoretical work on 1-swap? For example, could the authors derive a quantitative relationship between the number of 1-swap iterations and the reduction in pruning error, or prove the degree of approximation to a local optimum under specific conditions (such as row-level sparsity or SVD compression)?

---

> ### Author Response · Authors · 2025-11-21
>
> We thank you for your review and constructive feedback. We address your concerns below and hope that you reconsider your initial evaluation of our manuscript.
>
> ### Weaknesses
>
> > 1. The experiments in this paper are somewhat limited. Although results are provided for five LLM models, all of them are language models. It remains unclear how SparseSwaps performs on vision models or other types of Transformer architectures. This limitation constrains the generality and comprehensiveness of the evaluation.
>
> We focused entirely on language models (as e.g. indicated in the title), and the warmstart pruning methods considered are all tailored to LLMs. In general, we do not see any reason why SparseSwaps would not work for other types of Transformer architectures, with a minor exception: Sun et al. (2023) had found that the improvement of enforcing per-row instead of unstructured sparsity is a phenomenon that is specific to LLMs; so restricting to per-row sparsity could be problematic. For sparsity types that ensure the same level of sparsity anyways, such as N:M sparsity, the rows are by definition decoupled, and SparseSwaps will work out of the box. By design, SparseSwaps reduces the local pruning error compared to the baseline methods it uses as the warmstart (at least as long as beneficial 1-swaps are possible). Whether this reduction in local error pruning error is beneficial in terms of final performance depends on several factors, as we have shown in the paper. While we think that our focus on improving pruning methods for language models specifically is not too narrow, we are happy to provide results for other architectures in the camera-ready version if you think that this will significantly strengthen our paper.
>
> > 2. The paper does not provide the runtime of SparseSwaps on different models or comparisons with baselines, which makes it difficult to evaluate the proposed method.
>
> Thank you for the suggestion. By definition, mask refinement methods like SparseSwaps or DsNOT add computational overhead compared to the baseline methods they refine. That being said, SparseSwaps is a resource-intensive method (at least w.r.t. compute and our current implementation). We argue in the paper that the additional compute is worth it because it can be amortized over many inference requests, given the increased inference demands of LLMs. Furthermore, the additional compute is only linear in the number of 1-swap iterations $T_{\max}$, which is typically set to a small value. To address your concern, we have added the runtimes of LLaMA-3.1-8B on a single H100 GPU in the exact same setup as in Table 2 in the paper below (60% Sparsity), and will include a detailed discussion and analysis in the upcoming revision. Note that the case of 0 iterations corresponds to the full runtime of the method without doing any swaps, i.e., the sampling of calibration data, layerwise pruning using Wanda, including per-layer data collection and SVD computation for SparseSwaps, as well as the final Wikitext and EleutherAI evaluation. For reference, we have also computed the runtime of SparseGPT, which amounts to 35 minutes.
>
>
> | $T_{\text{max}}$ | 0 | 1 | 2 | 5 | 10 | 25 | 50 | 100 | 200 |
> | --- | --- | --- | --- | --- | --- | --- | --- | --- | --- |
> | **Walltime (hours)** | 2h45m | 2h54m | 3h13m | 3h18m | 3h35m | 4h01m | 4h51m | 6h22m | 10h03m |
>
>
> Some remarks:
> - We think that our implementation can be further improved w.r.t. efficiency. As the comparison between $T_{\max} = 0$ and $T_{\max} > 0$ as well as the per-transformer-block breakdown of a LLAMA-3.1-8B below (25 iterations, 50% sparsity) highlight, our implementation currently spends roughly half of the runtime on data operations; in part, this is due to an implementation where we do a single forward pass through each transformer block, save the necessary datafiles for each single layer matrix to the hard drive, and then load them back into memory when running the SparseSwaps algorithm for the layer. This is certainly a suboptimal implementation which we will improve for the camera-ready version and publication of the code.
> - Nevertheless, we acknowledge that SparseSwaps incurs non-trivial computational overhead. However, we emphasize that the algorithm is fully parallelizable across rows; one could solve each individual row on its own GPU if necessary.
> - Our main goal was to show that an greedy-exact approach like SparseSwaps is realizable; the efficiency can be drastically improved by relaxing the exactness requirement, and to allow e.g. doing more than one swap per iteration, and then only updating the cost vectors.

---

> > ### Author Response · Authors · 2025-11-21
> >
> > ```
> > ============================================================
> > BLOCK 0 TIMING SUMMARY
> > ============================================================
> > Data Collection:     88.485s (31.4%)
> > Matrix Processing:   193.336s (68.6%)
> >   - Data loading   : 42.357s (15.0%)
> >   - SVD Compression: 5.099s (1.8%)
> >   - SparseSwaps   : 142.532s (50.6%)
> >   - Error measurement: 0.079s (0.0%)
> >   - Cleanup        : 3.269s (1.2%)
> > BLOCK TOTAL:         281.821s (100.0%)
> > ============================================================
> > ```
> >
> > > 3. The paper lacks a theoretical or experimental characterization of the convergence behavior of 1-swap.
> >
> > In general, converging to a global optimum is NP-hard. However, there is a simple theoretical argument that shows that SparseSwaps will converge to an approximate 1-swap-local optimum; we describe it in full mathematical detail below to avoid any confusion.
> >
> > **Background:**
> > Let the row-wise pruning problem be defined by
> > $$
> > f(P) := \Vert \sum_{j\in P} S_{j,:} \Vert_2^2,
> > $$
> > where $P\subseteq[d]$ is the set of pruned indices.
> >
> > Define the improvement from a feasible swap $(p,u)$ as
> > $$
> > \Delta(p,u;P) := f(P) - f\big(P\setminus\{p\}\cup\{u\}\big).
> > $$
> >
> > We call $P$ a **$\varepsilon$-1-swap local optimum** if $\max_{(p,u)\text{ feasible}} \Delta(p,u;P) \le \varepsilon$. In other words, no single swap can reduce the loss by more than $\varepsilon$.
> >
> > We consider the SparseSwaps algorithm as described in the paper and assume in addition that it terminates when the maximal feasible improvement is smaller than $\varepsilon$, i.e. $\max_{(p,u)\text{ feasible}} \Delta(p,u;P_t) \le \varepsilon$.
> >
> > **Lemma.**
> >
> > Let $P_0$ be the initial mask. Then SparseSwaps performs at most $T \le \left\lceil \frac{f(P_0)}{\varepsilon} \right\rceil$ swap operations before reaching an $\varepsilon$-1-swap local optimum.
> >
> >  **Proof**
> >
> > Note that once SparseSwaps terminates, it has reached an $\varepsilon$-1-swap local optimum. Hence, we can assume that at iteration $t$, SparseSwaps does not terminate and performs a swap $(p_t,u_t)$. It follows that $\Delta_t := \Delta(p_t,u_t;P_t) > \varepsilon$ by definition of the stopping rule.
> >
> > By definition of $\Delta_t$, we have that $f(P_{t+1}) = f(P_t) - \Delta_t$. Since $\Delta_t > \varepsilon$, this yields the strict decrease $f(P_{t+1}) \le f(P_t) - \varepsilon$.
> >
> > Unrolling the recursion over $T$ executed swaps gives $f(P_T) \le f(P_0) - T\varepsilon$.
> >
> > Because $f(\cdot)$ is a squared Euclidean norm, it is nonnegative: $f(P_T) \ge 0$.
> >
> > Thus SparseSwaps can perform at most $T \le \left\lceil \frac{f(P_0)}{\varepsilon} \right\rceil$ iterations before termination. At that point, by definition of the stopping rule, $\max_{(p,u)\text{ feasible}} \Delta(p,u;P_T) \le \varepsilon$, i.e. $P_T$ is an $\varepsilon$-1-swap local optimum. $\square$
> >
> > We will consider adding a theoretical analysis for the camera-ready version; in general, we think that since the problem is NP-hard and SparseSwaps is a 1-swap based greedy algorithm, quantitative guarantees will very much depend on the assumptions we make about the individual matrices. We will get back to you in this regard.
> >
> > > 4. The experiments in the paper appear to use a fixed setup (see line 350), lacking evaluations of the method under different sequence lengths and on other datasets.
> >
> > Using a subset of C4 of fixed size and sequence length as the calibration dataset is common practice in the literature. As the warmstart masks also depend on that choice, we have not evaluated the method under different sequence lengths and on other datasets. We think that if the calibration dataset contains enough samples of sufficient length (implying that the warmstart masks are of somewhat reasonable quality), SparseSwaps will equally work. Note that we have included experiments with different number of calibration samples in Figure 2 in the appendix (discussed in the main part starting from line 450). We further ran an ablation study comparing Wanda and SparseSwaps when varying the sequence length while keeping the number of calibration samples fixed at 128. Both methods were equally impacted by the sequence length, which overall was rather minor in terms of final perplexity. We will add a detailed discussion of this to the camera-ready revision.

---

> > > ### Author Response · Authors · 2025-11-21
> > >
> > > ### Questions
> > >
> > > > 1. Can the authors provide a comparison of runtime and memory usage between SparseSwaps and other baselines, using the same experimental setup?
> > >
> > > Please see our answer above, we will include the full results in the upcoming revision. Memorywise, when processing each layer, we only need to keep the (d_in x d_in) matrix $S$ in memory. When performing the SVD, we never realize the full (d_in x B) matrix, but rather perform batched computation of $XX^T$ and compute the SVD by diagonalization of the resulting (d_in x d_in) matrix. In the individual iterations over the rows, we mainly need to keep the (d_in x d_in) matrix $S$ in memory, as well as the (d_in x 1) vector $s$ for the current cost vector. The practical implementation is a bit more convoluted and keeps additional buffers. We conducted some preliminary additional experiments to measure the peak memory requirements when pruning to 50\% sparsity for a LLAMA-3.1-8B on a single H100 GPU, and got the following results:
> > >
> > > - Wanda: 25590 MiB
> > > - SparseSwaps with Wanda warmstart: 49478 MiB
> > >
> > > > 2. Is the effectiveness of SparseSwaps limited to specific data distributions and experimental setups? Could the authors provide additional experimental results under different calibration data settings?
> > >
> > > We do not think that SparseSwaps is limited in that sense, as the warmstart mask would be equality affected, cf. our answer to the previous question and Figure 2 in the appendix.
> > >
> > > > 3. Since SparseSwaps depends on warm-start masks (such as outputs from Wanda or RIA), how sensitive is the method to the quality of these masks? If the warm-start mask is of low quality (e.g., randomly generated masks or poor heuristic pruning masks), can SparseSwaps still effectively minimize pruning errors?
> > >
> > > Yes, definitely; if the initial pruning mask error is high, SparseSwaps will lead to more drastic reductions. We highlighted this in Table 3 in the appendix, where we show drastic perplexity reduction for magnitude pruning as the warmstart. Our experiments consistently showed that the weaker the initial mask is, the more pronounced the effect of SparseSwaps. We will add further tables that juxtapose the reduction in pruning error for different warmstart methods. For the models in Table 3 and for 60% sparsity, we get the following average reduction in local pruning error, and hence consistently higher reductions given a weaker warmstart (and keeping the number of iterations fixed):
> > >
> > > | Warmstart vs. Avg. rel. error reduction (%)    | LLAMA-3.1-8B | Gemma-2-9B | DeepSeek-7B |
> > > |---------------------|:------------:|:----------:|:-----------:|
> > > | Magnitude Warmstart |   62.69%     |  58.13%    |   58.10%    |
> > > | Wanda Warmstart     |   43.29%     |  40.45%    |   36.96%    |
> > >
> > > > 4. Are there any plans to expand the theoretical work on 1-swap? For example, could the authors derive a quantitative relationship between the number of 1-swap iterations and the reduction in pruning error, or prove the degree of approximation to a local optimum under specific conditions (such as row-level sparsity or SVD compression)?
> > >
> > > Please see our answer above.
> > >
> > > Thanks again for your remarks, please let us know if further clarifications are needed.

---

### Author Response · Authors · 2025-12-03
**Revision: End of Discussion Period**

We would like to thank the reviewers again for their constructive feedback and for taking the time to review our manuscript. We have uploaded a major revision of the manuscript. Below, we provide a summary of the major changes for the reviewers and the AC; smaller changes and additions are incorporated but not listed here.

- We have revised the entire mathematical notation and derivation to rely on $G = XX^T$ instead of using the SVD of $X$ and working with the compressed representation $X'$. While this does not change any results, it simplifies the notation and derivation and makes them more accessible to readers. Furthermore, although SVD computation only made up a minor fraction of the runtime, we can now avoid it entirely by directly accumulating $G$ on-the-fly as calibration samples pass through the layer.
- Given this change, our argumentation in the abstract, introduction and contributions section has been revised to be more precise and to reflect the new formulation.
- As discussed, we have also included a runtime analysis of SparseSwaps on an H100 GPU in Table 2. Please note that the reported runtimes are *significantly* lower than the runtimes reported in the original rebuttal. This is because we *significantly* improved the efficiency of our implementation, removing a major bottleneck present previously. This includes adaptive batch sizes for the forward passes, on-the-fly accumulation of the Gram matrix instead of costly memory storage of the per-layer data matrices, and a much more efficient implementation of the SparseSwaps algorithm.

We believe that these major changes address the concerns raised by the reviewers. Thank you again for helping to improve our manuscript!

---

### Meta-Review · Area_Chair_QqHd · 2026-01-07

**Summary:**

This paper proposes SparseSwaps, a post-training pruning mask refinement method for large language models that makes the layer-wise mask optimization problem tractable by enforcing equal sparsity per row and applying an efficient exact 1-swap local search. The approach is technically well-motivated, supported by clear mathematical analysis, and empirically demonstrates consistent reductions in local pruning error, as well as improvements in perplexity and zero-shot accuracy over refinement baselines such as Wanda and DSnoT across several modern LLMs.

However, while the work is sound and carefully executed, reviewers raised concerns regarding the practical impact and generality of the method. In particular, the approach introduces substantial computational overhead, relies on restrictive sparsity assumptions, and does not consistently outperform alternative pruning frameworks that jointly optimize pruning and reconstruction. Overall, the paper represents a solid incremental contribution, but its practical significance and broader impact appear limited for an ICLR audience.

**Reviewer Concerns:**

Concerns addressed by the rebuttal:
1. The authors substantially improved the paper by adding detailed wall-clock runtime and memory usage analyses on modern hardware (H100 GPUs), including breakdowns across different numbers of swap iterations. These additions clarify the practical cost of SparseSwaps and address a major concern raised by multiple reviewers.
2. The rebuttal provides a clear argument that the method converges to an ε-1-swap local optimum in a finite number of steps. This resolves ambiguity regarding termination and monotonicity, even though global optimality remains intractable.
3. The authors convincingly demonstrate that SparseSwaps is not fragile to poor warm-starts and can yield larger relative gains when initialized from weaker pruning masks. This addresses concerns about sensitivity to initialization.

Concerns that remain outstanding:
1. Despite clear reductions in local pruning error, improvements in end-task metrics (perplexity and accuracy) are not consistent across sparsity levels, and in some settings are modest or absent. Moreover, alternative methods such as SparseGPT or ADMM-based pruning often achieve stronger results with significantly lower computational cost, raising questions about when SparseSwaps should be preferred in practice.
2. The tractability of SparseSwaps critically depends on enforcing equal sparsity per row (or N:M sparsity), which prevents reallocating sparsity across rows. While the authors acknowledge this limitation, it restricts the method’s applicability and may preclude globally better solutions under unstructured sparsity.
3. Although the authors argue that the additional computation can be amortized over inference, SparseSwaps remains substantially more expensive than one-shot pruning methods. The current implementation and design suggest that, in many realistic scenarios, the performance gains may not justify the added complexity.

**Reviewer Scores:**

Reviewer svrG: Reviewer svrG would likely increase their score slightly after the added runtime analysis and theoretical clarification, but their concerns regarding limited evaluation scope and practical usefulness remain, keeping the score near the acceptance threshold.
Reviewer ohzU: Reviewer ohzU acknowledged the technical strengths of the method but remained concerned about computational overhead and limited ablation, and would likely maintain a marginally positive but cautious score.
Reviewer Q6G3: Reviewer Q6G3 was strongly positive throughout, but represents a minority view emphasizing technical novelty over practical impact.
Reviewer 1jXE: Reviewer 1jXE maintained a negative assessment, citing limited impact relative to existing pruning frameworks and unfavorable cost–performance trade-offs.

---

### Decision · Program_Chairs · 2026-01-26

Reject